# A Cross-Tissue Investigation of Molecular Targets and Physiological Functions of Nsun6 Using Knockout Mice

**DOI:** 10.3390/ijms23126584

**Published:** 2022-06-13

**Authors:** Wen Wang, Hengjun Huang, Hao Jiang, Chi Tian, Yisen Tang, Diwen Gan, Xiaozhen Wen, Zhenyu Song, Yuhao He, Xijun Ou, Liang Fang

**Affiliations:** 1Harbin Institute of Technology, Harbin 150001, China; buyushizheng@outlook.com; 2Department of Biology, Southern University of Science and Technology, Shenzhen 518005, China; 11930740@mail.sustech.edu.cn (H.H.); john-jh@foxmail.com (H.J.); 11930659@mail.sustech.edu.cn (C.T.); 11510575@mail.sustech.edu.cn (Y.T.); 11812416@mail.sustech.edu.cn (D.G.); 12133067@mail.sustech.edu.cn (X.W.); 11811113@mail.sustech.edu.cn (Z.S.); heyh@mail.sustech.edu.cn (Y.H.); ouxj@sustech.edu.cn (X.O.); 3Academy for Advanced Interdisciplinary Studies, Southern University of Science and Technology, Shenzhen 518005, China

**Keywords:** Nsun6, 5-methylcytosine, knockout mouse, phenotype

## Abstract

The 5-methylcytosine (m5C) modification on an mRNA molecule is deposited by Nsun2 and its paralog Nsun6. While the physiological functions of Nsun2 have been carefully studied using gene knockout (KO) mice, the physiological functions of Nsun6 remain elusive. In this study, we generated an Nsun6-KO mouse strain, which exhibited no apparent phenotype in both the development and adult stages as compared to wild-type mice. Taking advantage of this mouse strain, we identified 80 high-confident Nsun6-dependent m5C sites by mRNA bisulfite sequencing in five different tissues and systematically analyzed the transcriptomic phenotypes of Nsun6-KO tissues by mRNA sequencing. Our data indicated that Nsun6 is not required for the homeostasis of these organs under laboratory housing conditions, but its loss may affect immune response in the spleen and oxidoreductive reaction in the liver under certain conditions. Additionally, we further investigated T-cell-dependent B cell activation in KO mice and found that Nsun6 is not essential for the germinal center B cell formation but is associated with the formation of antibody-secreting plasma cells. Finally, we found that Nsun6-mediated m5C modification does not have any evident influence on the stability of Nsun6 target mRNAs, suggesting that Nsun6-KO-induced phenotypes may be associated with other functions of the m5C modification or Nsun6 protein.

## 1. Introduction

RNA modification plays a critical role in RNA metabolism and is conserved from prokaryotes to eukaryotes [1]. 5-methylcytosine (m5C) is one of the prevalent RNA modifications widely existing in different RNA species (rRNA, tRNA, and mRNA) [2]. Similar to N6-methyladenosine (m6A), the m5C modification is dynamic and reversible. It is catalyzed by the NSUN family (NSUN1-7) [3,4,5,6] and DNMT2 [7], demethylated by the TET family [8] and ALKBH1 [9], and recognized by reader proteins ALYREF [10] and YBX1 [11].

Among these m5C regulators, NSUN2 is the most widely studied due to its broad target spectrum and critical biological functions. It methylates positions C34, 48, 49, and 50 of tRNAs [12] and a large number of mRNA sites [13]. NSUN2 improves tRNA stability by preventing its cleavage by Angiogenin under stress conditions [12,14]. It also facilitates the nuclear export and improves the stability of mRNA through reader proteins ALYREF and YBX1, respectively [10,11]. The knockout (KO) of Nsun2 causes weight loss, partial alopecia, and meiosis abnormality in male mice [15,16].

NSUN6 has been known to mediate the m5C modification on position C72 of tRNA^Cys^ and tRNA^Thr^ since 2015 [17,18,19]. Until recently, we and other groups have reported that NSUN6 is responsible for many mRNA m5C sites in different human cell lines that had not been modified by NSUN2 [20,21,22], indicating that the previously reported functions of the m5C modification on mRNA may be partially explained by NSUN6. The pathological functions of NSUN6 have been reported in the context of human cancer: the overexpression of NSUN6 repressed the proliferation of liver cancer cells [23], and the knockdown of NSUN6 repressed the migration of breast cancer cells [24]. However, the physiological functions of Nsun6 have not been systematically studied using mouse genetics.

Here, we constructed an Nsun6-KO mouse strain, which exhibited no apparent phenotypes in both the development and adult stages as compared to wild-type mice. Through mRNA bisulfite sequencing (BS-seq) in wild-type (WT) as well as KO mice, we identified 609 m5C sites in five different tissues, 80 of which were Nsun6-dependent. The molecular influence of Nsun6 KO was analyzed by mRNA sequencing (RNA-seq) in these tissues, which demonstrated that Nsun6 is not required for organ homeostasis under laboratory housing conditions, but its loss may affect immune response in the spleen and oxidoreductive reaction in the liver under certain conditions. Additionally, we further investigated T-cell-dependent B cell activation in KO mice and found that Nsun6 is not essential for the germinal center (GC) B cell formation but is associated with the formation of antibody-secreting plasma cells. Furthermore, our data demonstrated that the Nsun6-mediated m5C modification did not have any evident influence on the stability of Nsun6 target mRNAs, suggesting that the Nsun6-KO-induced molecular phenotypes may be associated with other functions of the m5C modification or Nsun6 protein.

## 2. Results

### 2.1. Nsun6 Gene Is Not Essential for the Development and Breeding of Mice

To generate the Nsun6-KO mouse strain, we used the CRISPR/Cas9 system to delete the exon 3 and 4 of Nsun6 with two single-guide RNAs (sgRNAs) targeting intron 2 and 4, respectively (Figure 1A). The expected genotype was determined by PCR (Figure 1B), while the protein-level depletion was confirmed by Western blot (Appendix A). Previous studies reported that Nsun2-KO male mice showed phenotypes, including weight loss, partial alopecia, and meiosis abnormality [15,16]. Given that Nsun6 is also responsible for the m5C modification of many mRNAs, we speculate that Nsun6 KO may also cause certain phenotypes under physiological conditions. Surprisingly, Nsun6-KO mice developed normally and showed normal appearance and weight in adults as compared with wild-type (WT) mice (*n* = 10, *p*-value > 0.5) (Figure 1C,D). Both female and male Nsun6-KO mice were also fertile and gave birth to healthy Nsun6-KO offspring. In summary, even though Nsun6 has similar molecular functions as Nsun2, the physiological consequence of Nsun6 KO is quite different, indicating that the m5C modification mediated by Nsun6 may have very different molecular functions.

### 2.2. Identification of High-Confident Nsun6-Dependent m5C Sites across Tissues

Previous studies demonstrated that the m5C modification is associated with mRNA metabolism, including nuclear export, degradation, and translation. To investigate the potential molecular functions of Nsun6, we first sought to identify Nsun6-mediated m5C modification on the mRNA. Although some studies have reported the m5C methylome of mouse tissues [10,25], the question as to which of these sites had been edited by Nsun6 remains unclear. Therefore, we performed BS-seq for five different tissues (small intestine, liver, spleen, kidney, and heart) of WT as well as Nsun6-KO mice, respectively (Figure 2A). After initial processing, 30–58 million read pairs were obtained from BS-seq for each sample (Appendix A), and the bisulfate conversion rate in all samples was above 98% (Appendix A).

Several methods have been developed to identify m5C sites from BS-seq data. Here, we chose the pipeline developed by Huang et al. [25] due to its low false-positive detection rate, and m5C sites with a methylation level >0.05 in WT samples and =0 in KO samples were defined as high-confident Nsun6-dependent m5C sites (Appendix A). In total, we identified 119, 97, 113, 170, and 409 m5C sites in the small intestine, liver, spleen, kidney, and heart samples, of which 18, 14, 7, 16, and 39 were Nsun6-dependent, respectively (Figure 2B and Appendix A). Unexpectedly, the proportion of the Nsun6-dependent m5C sites in mouse tissues is much lower than that in the human cell line [20]. Then, we took the union of all m5C sites detected in the five tissues, which included a total of 639 m5C sites and 80 Nsun6-dependent ones, in order to analyze the features of Nsun6-dependent m5C sites. Compared to all m5C sites, which were enriched near the start and stop codon, Nsun6-dependent m5C sites were only enriched near the stop codon (Figure 2C), which agrees with what has been observed in human cell lines [22]. Subsequently, the motif analysis was performed with Nsun6-dependent m5C sites to illustrate the sequence preference of Nsun6 in mice. As shown in Appendix A, Nsun6 had a sequence preference of “m5CTCCA”, which is consistent with our previously reported NSUN6-dependent m5C motif in the human cell line [20], suggesting the conversed target motif of Nsun6. However, these m5C sites are not conserved at the gene level. For instance, although the genes FURIN and NECTIN2 are highly modified m5Cs in human cell lines (Fang et al., 2020), their orthologous genes Furin and Nectin2 were not methylated in mouse tissue, where their expression could be detected (Appendix A). Vice versa, the Igfbp6 gene was highly methylated in mouse small intestine and heart, but its ortholog IGFBP6 was not methylated in human cells [20]. These results indicate that the m5C sites in mice and humans were conserved in their local sequences but not in the genes.

To further explore the features of Nsun6-dependent m5C sites, we plotted the modification rate of Nsun6-dependent m5C sites across tissues (Figure 2D). As shown, most genes showed tissue-specific modification due to two major reasons: the tissue-specific expression of the gene and tissue-specific targeting by Nsun6. For example, the site in Igfbp6, which was highly methylated in the small intestine and heart, belongs to the first category; the site in the gene Vwa8, which was methylated in the liver but not in the small intestine and kidney, belongs to the second category. The presence of genes such as Vwa8 suggests that Nsun6-mediated m5C modifications are under fine regulation.

Interestingly, compared to other tissues, the heart had relatively more m5C sites. To check if this is due to the higher protein level of Nsun6 in the heart, Western blot was performed, which revealed that the Nsun6 protein level in the heart was relatively low (Appendix A). To further investigate the potential functions of Nsun6 target genes in the heart, Gene Ontology (GO) analysis was performed. While all genes with m5C sites were found to be enriched for molecules involved in “organelle inner member” (Appendix A), genes with Nsun6-dependent sites were specifically enriched for molecules associated with “anchoring junction” and “focal adhesion” (Figure 2E), suggesting that Nsun6 may be involved in the regulation of cell–matrix interactions in the heart.

### 2.3. Nsun6 Deficiency Induces Limited Transcriptome Changes across Tissues

Given that Nsun6-KO mice did not show abnormal appearance, we investigated if Nsun6 KO could induce any molecular phenotype in the mouse. The same five tissues were chosen to perform RNA-seq with two biological replicates of the WT and KO samples. Among these samples, 18–35 million cleaned reads were obtained and then mapped to a mouse reference genome (mm10) with a mapping rate of about 95%. Finally, gene expression levels were determined by counting each gene coverage with ENSEMBL GRCm38 annotation for further analysis. The *t*-SNE analysis revealed that the samples are well-clustered according to the tissue origin (Figure 3A).

The DESeq was utilized to identify the differentially expressed genes (DEGs) between WT and KO tissues within each tissue group (FDR < 0.1, |FoldChange| > 2). There were 27, 33, 200, 0, and 0 upregulated genes, and 13, 20, 8, 0, and 10 downregulated genes in the small intestine, liver, spleen, kidney, and heart of KO mice, respectively (Figure 3B and Appendix A), which suggests that the effect of Nsun6 KO in the spleen is relatively stronger than in other tissues. The Venn diagram of the DEGs among different tissue suggests that most DEGs are tissue-specific: only two genes, Isg15 and E2f8, are upregulated in both the liver and spleen, and only one gene, Gm47283, is downregulated in both the liver and spleen (Figure 3C).

To study the potential biological functions of Nsun6 in mice, tissues (liver and spleen) with relatively more DEGs were chosen for gene functional enrichment analysis. In the liver, the GO enrichment analysis showed that upregulated genes were enriched in “oxidoreductase activity”, while downregulated genes were enriched in the “fatty acid metabolic process” (Appendix A). In the spleen, upregulated genes were enriched in the GO pathways of “chromosome segregation”, “erythrocyte homeostasis”, and “myeloid cell homeostasis”, while downregulated genes were significantly enriched in “immunoglobulin production” and “production of molecular mediator of immune response” (Figure 3D). Taken together, these results suggest that although Nsun6 is not essential for the homeostasis of these five organs under laboratory housing conditions, it may be involved in regulating myeloid cell homeostasis and antibody secretion in the spleen and oxidoreductive reaction in the liver under certain conditions.

### 2.4. Nsun6-KO Impairs the Formation of Antibody-Secreting Plasma Cells in the Spleen

Based on the GO analysis of the DEGs in the spleen, we sought to examine in detail whether Nsun6 deficiency could indeed cause a biological phenotype in the spleen under homeostasis and perturbation. Given that the spleen is the largest secondary organ in the immune system and a major site of immune response to antigens [26], we first examined the development of various types of immune cells in the Nsun6 KO spleen and found that KO mice had a normal spleen size (Appendix A) and a comparable proportion of B220^+^ B cells, F4/80^+^ monocytes/macrophages, Gr1^+^ neutrophils, B220^−^NK1.1^+^ NK cells, as well as CD4^+^CD8^-^ and CD4^−^CD8^+^ T-cells between WT and KO mice (Appendix A), indicating that Nsun6 is dispensable in immune cell development in the spleen.

A previous study has shown that Nsun6 is mainly located in the Golgi apparatus [17], whose main function is to process, sort, and transport proteins synthesized in the endoplasmic reticulum and then distribute these proteins to specific parts of the cell or directly secrete them outside the cell. These suggest that Nsun6 may perform specific functions in immune response or antibody secretion. We therefore examined the function of Nsun6 in antibody immune response by challenging WT and Nsun6 KO mice with 4-Hydroxy-3-nitrophenylacetyl-Chicken Gamma Globulin (NP-CGG) in alum to induce T-cell-dependent antibody immune responses. We assessed activated B cells in the spleen of WT and KO mice by flow cytometry 10 days post-immunization. Germinal center (GC) B cells are the precursor cells that differentiate into long-lasting plasma cells and memory B cells [27]. We found that the proportion and the number of B220^+^Fas^+^CD38^-^ GC B cells were largely normal in the spleen of KO mice as compared to that of the WT mice (Figure 4A,B). In addition, the ratio of CXCR4^high(hi)^ CD83^low(lo)^ DZ and CXCR4^lo^CD83^hi^ LZ B cells in the GC B cell compartment was also comparable between WT and KO mice (Figure 4C,D). Consistently, the percentage and number of NP-specific IgG1^+^NIP^+^ B cells were unchanged after Nsun6 deletion (Figure 4E,F), indicating that Nsun6 is not essential for GC B cell formation. Interestingly, the ELISPOT analysis of antibody-secreting cells (ASCs) showed that the number of NP-specific IgG1 ASCs in the spleen of KO mice was significantly reduced by half as compared with WT mice (Figure 4G,H). In summary, these results suggest that Nsun6 is dispensable in GC B cell development but plays a regulatory role in the formation of antibody-secreting plasma cells.

### 2.5. Nsun6-Mediated m5C Modification Does Not Affect RNA Stability across Tissues

Previous studies have reported that m5C is involved in RNA stability [11,28]. Therefore, we investigated if Nsun6-KO would affect the mRNA abundance of its target genes. The expression levels of Nsun6 target genes were compared between WT and KO samples, which revealed that in all tissues, Nsun6 target genes showed both mild upregulation and downregulation in the Nsun6-KO samples, and in general, the genes were evenly distributed on both sides of the diagonal line (Figure 5). Additionally, no global difference could be observed between the genes with Nsun6-dependent and -independent sites (Figure 6A).

Given that the methylation levels of different genes vary significantly, and that genes with low modification rates may show limited changes upon Nsun6 KO, we selected genes with high modification rates in the heart for further analysis using RT-qPCR. We found that the mRNA levels of Igfbp6, Afg3l1, and Csf1r presented no significant changes between WT and KO heart samples (Figure 6B). These results suggest that in the five tissues we analyzed, Nsun6-mediated m5C modification does not have any evident influence on RNA stability, and Nsun6 KO-induced phenotypes may be associated with other functions of the m5C modification or with Nsun6 itself.

## 3. Discussion

5-methylcytosine (m5C), one of the prevalent RNA modifications, is deposited by NSUN2 and its paralog NSUN6 on an mRNA [20,21,22]. While the physiological functions of Nsun2 have been deeply studied using the gene KO mouse, the physiological functions of Nsun6 remain elusive. Here, we generated an Nsun6-KO mouse strain, which helped to identify high-confident Nsun6-dependent m5C sites in five different tissues. The molecular influence of Nsun6 KO in these tissues was analyzed by RNA-seq, which demonstrated that Nsun6 deficiency induces limited transcriptome changes across tissues. Due to the relatively larger number of DEGs detected in the KO spleen, we further investigated its function in the spleen and found that Nsun6 is not essential for the GC B cell development but is involved in the regulation of ASCs.

It has been shown that the loss of Nsun2 could cause several apparent phenotypes in male mice, including weight loss, partial alopecia, and meiosis abnormality [15,16]. Given that Nsun6 has a similar molecular function and is highly conserved from archaea to humans [19], we expected Nsun6 to be critical for certain biological processes. Therefore, it is surprising that Nsun6 deficiency did not induce any Nsun2-related or other apparent phenotypes. Further DEG analysis also demonstrated that the loss of Nsun6 only resulted in limited transcriptome changes across five tissues with only three overlapped DEGs. This demonstrated that Nsun6 is not required for the development and homeostasis of the animal, but also indicates that it may be involved in more elaborate regulation, or its function is associated with specific perturbation (e.g., stress). As demonstrated in the investigation of antibody immune response in the KO spleen, we indeed found that Nsun6 is dispensable in the development of the B cell but is associated with the formation of ASCs under an immune challenge. Given that Nsun6 is mainly located in the Golgi apparatus [17], whose main function is to process, sort, and transport proteins synthesized in the endoplasmic reticulum and to distribute them to specific parts of the cell or directly secrete them outside the cell, we speculate that Nsun6 may affect the metabolism of antibody proteins. However, how Nsun6 affects the formation of ASCs at the molecular level, and whether Nsun6 is involved in other biological processes requires a more detailed analysis of the KO mice.

Interestingly, although the heart has relatively more m5C sites, it does not have the highest protein level of Nsun6, indicating that the m5C modification may be globally regulated by certain factors. Whether these co-factors that enhance the activity of Nsun6 exist and whether the concentration of S-adenosine methionine (SAM), the donor of the methyl group, could affect the modification rate are both worth further investigations.

It has been reported that m5C methylation in mRNAs could affect RNA stability, RNA location, and translation efficiency [10,25,28,29]. However, our RNA-seq data showed that the RNA levels of genes with Nsun6-dependent m5C sites did not significantly change after Nsun6-KO. Taking advantage of the NSUN6-KO HAP1 cells constructed in our previous study [20], we separated nuclear and cytoplasmic fractions of RNA for analysis and found that the distribution of NSUN6-modified mRNAs was not altered as compared to WT cells (data not shown). Additionally, through polysome profiling followed by RNA-seq, we observed that there were no differences in the translation efficiency of NSUN6-modified mRNAs between WT and KO HAP1 cells (data not shown). These could potentially be explained by two reasons: (1) the mRNA stability regulated by m5C may only be observed in specific physiological conditions (e.g., the maternal to zygotic transition) due to the presence of specific m5C readers, and (2) m5C readers involved in mRNA metabolism may only recognize Nsun2-dependent m5C sites due to the different local sequence feature. Therefore, how NSUN6-mediated m5C modification contributes to mRNA metabolism in different contexts remains elusive, and the Nsun6-KO mouse strain is an important resource to tackle these questions.

In this study, we generated an Nsun6-KO mouse strain and investigated the molecular phenotypes of five different organs with systematic approaches. Although the limited influence of Nsun6 deficiency was observed, we uncovered its involvement in antibody immune response in the spleen. Our work provides a deeper understanding of Nsun6′s phycological functions and paves the way for further Nsun6-related studies.

## 4. Methods

### 4.1. Generation of the Nsun6-Knockout Mouse Strain

The Nsun6-KO mouse was generated by the CRISPR/Cas9 system. sgRNAs were transcribed in vitro and microinjected with the Cas9 protein into the fertilized eggs of C57BL/6J mice. Fertilized eggs were transplanted to obtain positive F0 mice. The exon 3 and exon 4 were removed by sgRNAs: CCTGTAAAGAAGTCCGAGCT and TTCCTGACTCGAGCCAGTAA. The genotype was confirmed by the following primers: F1, GAGTGCTAAGATTACAGATGCACACC; R1, CTATCTAAATCTTCAACAGGAGGCATC; F2, GATCCTGTCCGAGGATGATGAGC; R2, CCTGAATTCGATCCACATGGGTC. Homologous Nsun6 mice were obtained by mating heterozygous Nsun6 mice.

### 4.2. Total RNA Extraction

The mice were sacrificed by cervical dislocation. Tissues, including small intestine, liver, spleen, kidney, and heart, were obtained from 8-week-old female mice. Tissues were dissected, snap-frozen by liquid nitrogen, and kept at −80 °C. All tissues were ground by mortar before RNA extraction. Total RNA was extracted by TRIzol^®^ Reagent Invitrogen, #15596026, Carlsbad, CA, USA) with DNase I treatment according to the manufacturer’s instructions.

### 4.3. mRNA Capture

mRNA was captured by VAHTS mRNA Capture Beads (Vazyme, #N401,Nanjing, China) according to the manufacturer’s instructions.

### 4.4. mRNA Library Construction

A strand-specific mRNA library was constructed by VAHTS^®^ Stranded mRNA-seq Library Prep Kit for Illumina^®^ V2 (Vazyme, NR612, Nanjing, China) according to the manufacturer’s instructions.

### 4.5. mRNA Bisulfite Sequencing

A total of 200 ng mRNA was converted by EZ RNA methylation kit (Zymo Research, CA, USA) according to the manufacturer’s instructions, with minor modifications. Briefly, the mRNA was incubated at 70 °C for 30 min and at 60 °C for 1 h at the bisulfite treatment step. The quality of the bisulfite-treated mRNA was assessed using Agilent RNA 6000 Pico Kit (Agilent Technologies, Santa Clara, CA, USA), and it was then applied in the preparation of NGS libraries using the VAHTS Stranded mRNA-seq Library Prep Kit (Vazyme, Nanjing, China). The library quality was assessed using the High Sensitivity DNA Kit (Agilent Technologies, Santa Clara, CA, USA). Paired-end sequencing (2 × 150 bp) was performed at the HaploX Genomics Center (HaploX Biotechnology Co., Ltd., Shenzhen, China).

### 4.6. Western Blot

All tissues were ground by mortar before protein extraction. Total protein was extracted by RIPA lysis buffer (Beyotime, P0013B, Shanghai, China) and sonicated for 10 cycles (30s On/30s Off). Protein concentration was measured by BCA (Beyotime, P0011, Shanghai, China). A total of 60 µg of protein was loaded for Western blot. The following antibodies were used in this study: Nsun6 (Abclonal, A7205, Wuhan, China) and Gapdh (Proteintech, 60004-1-lg, Wuhan, China).

### 4.7. Detection of the m5C Modification

The m5C sites on the mRNA were detected by the RNA-m5C pipeline according to our previous study [20].

### 4.8. Nsun6-Dependent Site Identification

In this study, the criteria for determining the high-confident m5C sites were as follows: (1) the coverage of the site should have at least 20 reads in 2 replicates; (2) the number of reads containing the unmodified C should be at least 2 in 2 replicates; (3) the WT methylation level should be at least 0.05.

### 4.9. Mice Immunization

The mice were intraperitoneally immunized with 100 μg NP37-CGG (Biosearch Technologies, Novato, CA, USA) in aluminum (Thermo, Waltham, MA, USA) for cell-dependent humoral immune response.

### 4.10. Flow Cytometry

Single-cell suspensions were prepared from mouse spleens by the removal of red cells using an RBC lysis buffer (BioLegend, San Diego, CA, USA) and by passing them through a 70 µm nylon mesh. After counting, the cells were stained with the indicated antibodies: anti-B220-APC-Cy7 (BD Biosciences, San Diego, CA, USA, RA3-6B2), CD38-PerCP-Cy5.5 (BD Biosciences, San Diego, CA, USA, 90/CD38), FAS-PE (eBiosciences, San Diego, CA, USA, 15A7), CXCR4-APC (eBiosciences, San Diego, CA, USA, 2B11), CD83-FITC (Invitrogen, Carlsbad, CA, USA, Michel-17), IgG1-FITC (BD Biosciences, San Diego, CA, USA, A85-1), Dump (CD138-biotin (BD Biosciences, San Diego, USA, 281-2), IgM-biotin (BD Biosciences, San Diego, CA,USA, R6-60.2), IgD-biotin (BD Biosciences, San Diego, CA, USA, 217-170), Gr-1-biotin (eBiosciences, San Diego, CA, USA, RB6-8C5), CD3-biotin (BioLegend, San Diego, CA, USA, 17A2)) conjugated with PerCP Streptavidin (BD Biosciences, San Diego, USA, 554064), F4/80-PE (eBiosciences, San Diego,CA, USA, BM8), Gr-1-FITC (eBiosciences, San Diego, USA, RB6-8C5), NK1.1-biotin (BioLegend, San Diego, CA, USA, PK136) conjugated with PE-Cy7 Streptavidin (BD Biosciences, San Diego,CA, USA, 557598), CD4-APC (BD Biosciences, San Diego, CA, USA, RM4-5), and CD8a-PerCP-Cy5.5 (eBiosciences, San Diego, CA, USA, 53-6.7). NP-specific B cells were detected by NIP-BSA (Biosearch Technologies, Novato, CA, USA) conjugated with PE using R-Phycoerythrin Labeling Kit-SH (Dojindo, Kumamoto, Japan). Data were analyzed using Flowjo V10.7

### 4.11. ELISPOT Assay

One million splenocytes were seeded in each well of the MultiScreen filter plate (Milipore, Darmstadt, Germany) coated with NP20-BSA (Biosearch Technologies, Novato, CA, USA) and blocked with BSA for incubation. After washing, the plate was incubated with biotin-conjugated anti-IgG1 antibodies (1/1000, SouthernBiotech, Birmingham, AL, USA) for 1 h at room temperature, and with streptavidin-AP (1/1000, SouthernBiotech, Birmingham, AL, USA) for another 0.5 h. Then, the antibody-secreting cells were visualized using substrates (Mabtech, Cincinnati, OH, USA).

### 4.12. RNA-seq Analysis

Raw pair-end reads were trimmed by cutadapt (-a AGATCGGAAGAG-A AGATCGGAAGAG-j 8). Cleaned reads were mapped to mouse reference genomes (GRCh38) by hisat2 (default parameters). All mapping results were counted by featureCounts (-t exon -s 2 -B -C, annotation file is GRCm38.102). The DEGs were calculated by DEseq2 (fc > 2, fdr < 0.1).

### 4.13. t-SNE Visualization

t-SNE visualization was performed using the Package Rtsne version 0.15 of the R Package.

### 4.14. GO Enrichment Analysis

GO enrichment was performed using the clusterProfiler of the R package. For each tissue type, the expressed genes were used as background.

### 4.15. GSEA Analysis

Genes with a TPM larger than 1 in at least one sample were used to perform GSEA analysis using the GSEA software version 4.1.0 with default parameters, except that the metric for ranking genes was the log2 Ratio of classes.

### 4.16. Motif Analysis of m5C Sites

The upstream and downstream 10 nt sequences flanking the m5C sites were extracted from the genome and subjected to ggseqlogo [30] for motif analysis.

### 4.17. Data Availability

All sequencing data are available in the GEO database under accession number GSE200953.

## Figures and Tables

**Figure 1 ijms-23-06584-f001:**
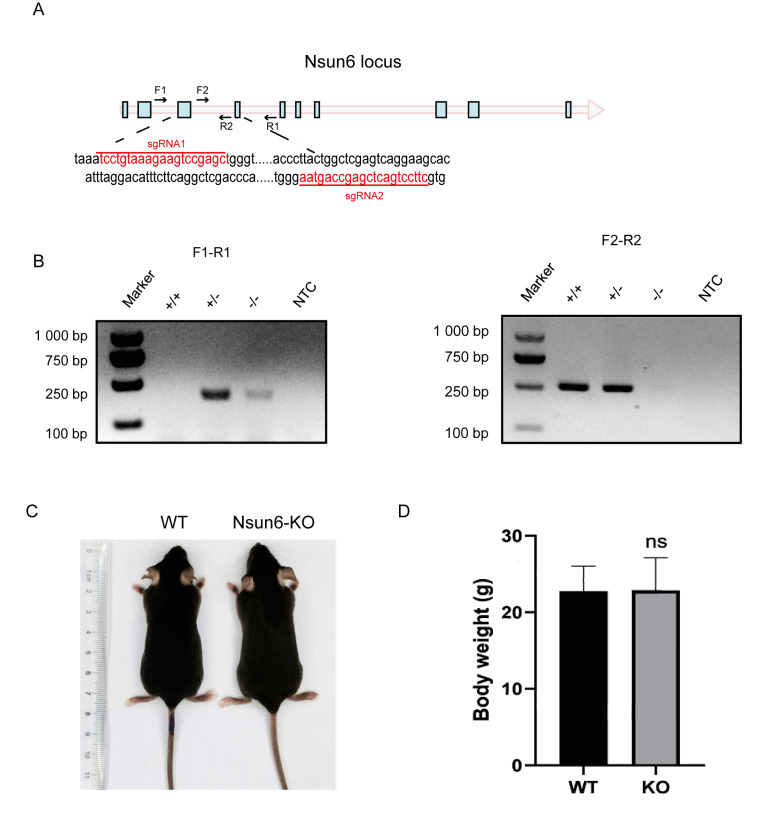
Construction of the Nsun6-KO mouse strain. (**A**) The schematic design of the CRISPR/Cas-mediated Nsun6 KO, where two sgRNAs were used to remove exon 3 and 4 of the gene. Two PCR primer pairs were designed to confirm the genotype. (**B**) The correct genotype of the KO mouse was confirmed: the primer pairs F1-R1 and F2-R2 were used to detect the KO and WT allele, respectively. (**C**) The appearance of Nsun6-KO and WT mice. (**D**) The bodyweight of Nsun6-KO and WT mice at the age of 8 weeks (*n* = 5, *p* > 0.5).

**Figure 2 ijms-23-06584-f002:**
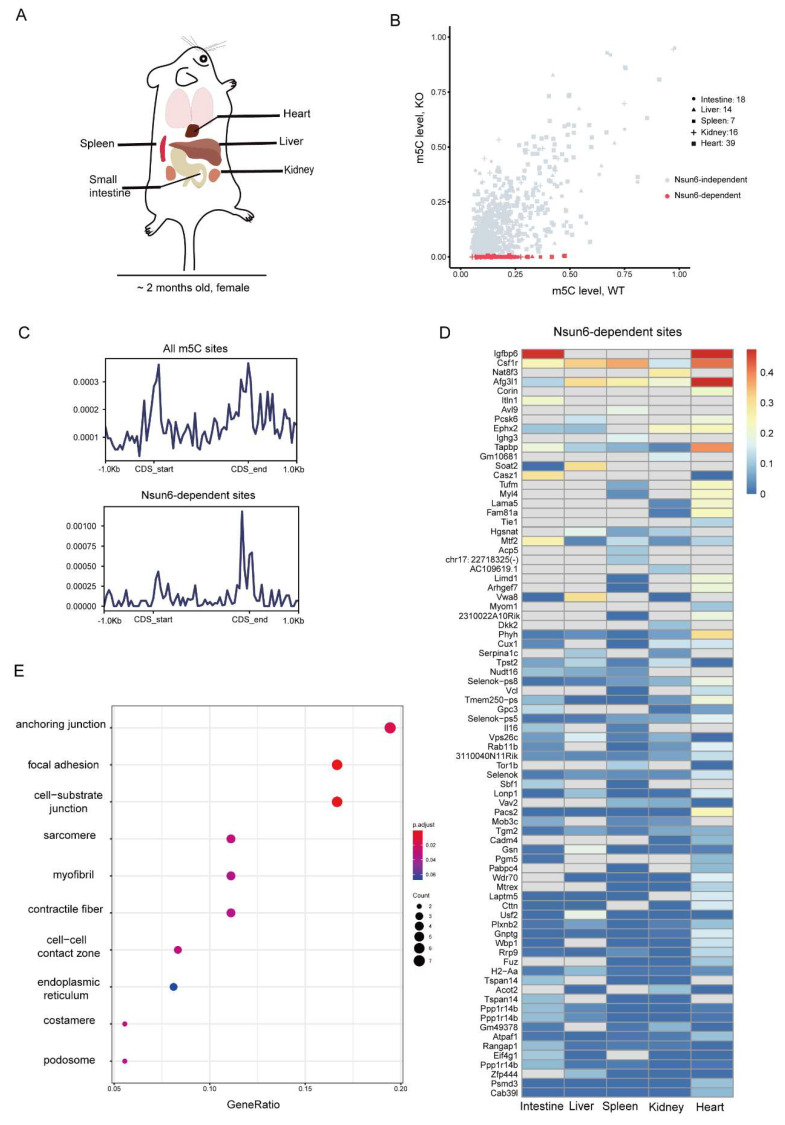
Identification of high-confident Nsun6-dependent m5C sites across tissues. (**A**) Five tissues (small intestine, liver, spleen, kidney, and heart) were used to perform BS-seq (*n* = 2). (**B**) The scatter plot presents the modification rate of m5C sites in different tissues. Red dots represent Nsun6-dependent sites. (**C**) The distribution of total and Nsun6-dependent m5C sites along with mRNA. (**D**) Heat map showing the modification rate of Nsun6-dependent sites across tissues. Undetected sites were represented by the gray block. (**E**) GO analysis of genes with Nsun6-dependent m5C sites in the heart tissue.

**Figure 3 ijms-23-06584-f003:**
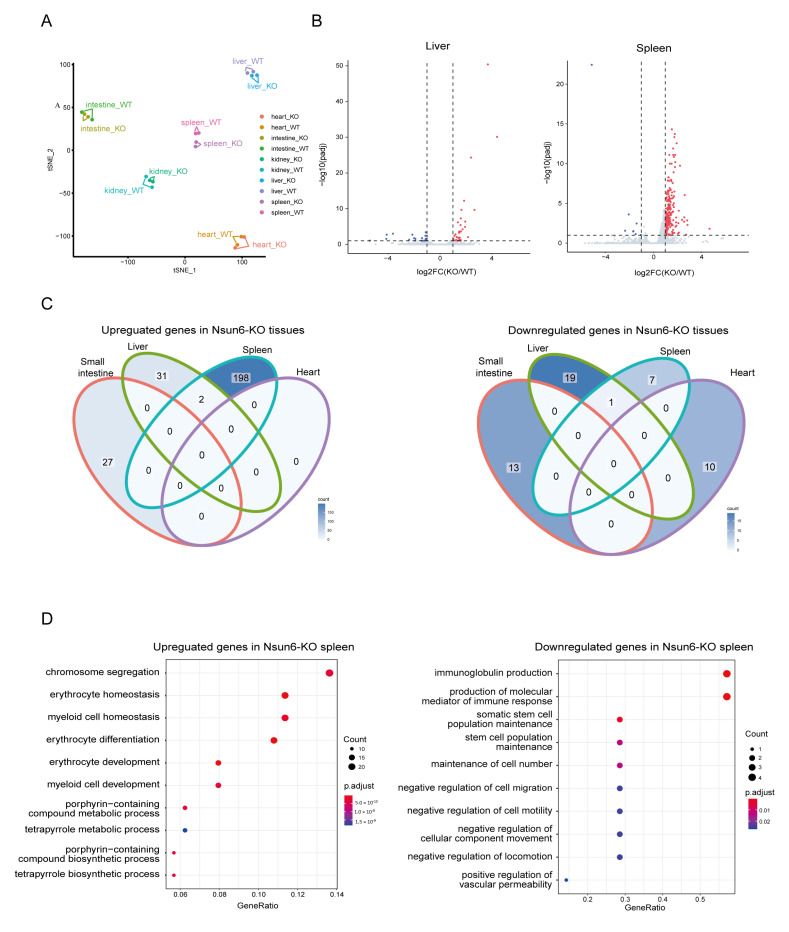
The influence of Nsun6 KO on transcriptome across tissues. (**A**) A t-SNE plot presenting the transcriptomic similarity and divergence of all RNA-seq data sets. (**B**) Volcano plots presenting the DEGs identified in the liver and spleen of KO mice. (**C**) A Venn diagram of the DEGs in Nsun6-KO small intestine, liver, spleen, and heart. (**D**) GO analysis of the DEGs in Nsun6-KO spleen.

**Figure 4 ijms-23-06584-f004:**
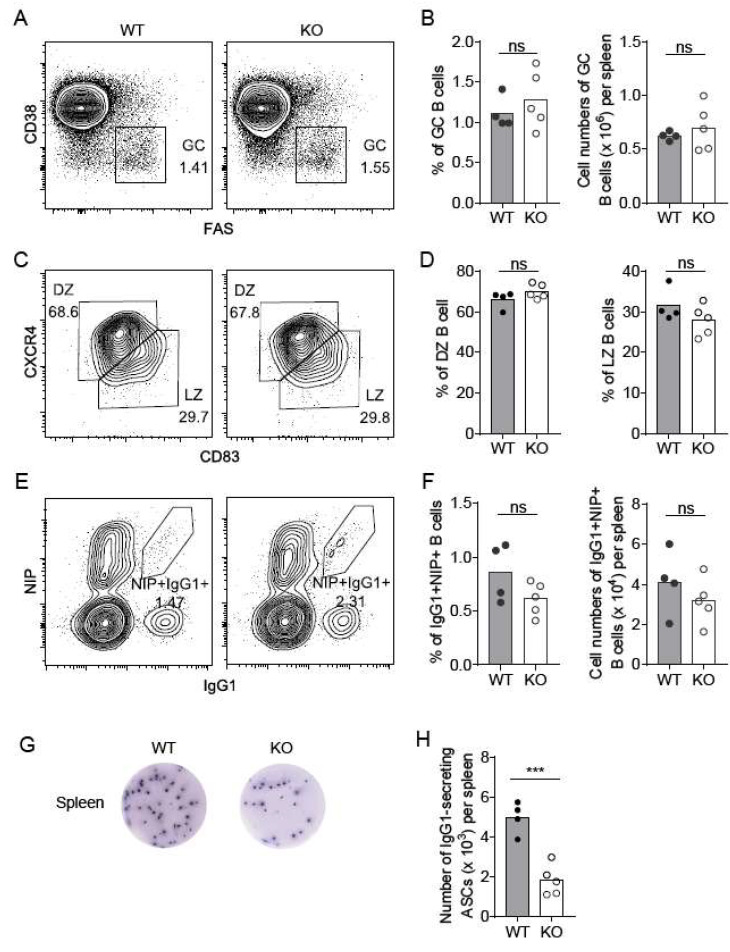
Nsun6 is dispensable in GC B cell response but important for antigen-specific ASC formation. (**A**) Flow cytometric analysis of the GC B cell (B220^+^CD38^-^FAS^+^) in the spleen of WT and KO mice. (**B**) The ratio and number of GC B cells. (**C**) Flow cytometric analysis of DZ (B220^+^ CD38^-^FAS^+^CXCR4^high^CD83^low^) and LZ (B220^+^CD38^-^FAS^+^CXCR4^low^CD83^high^) B cells. (**D**) The proportion of DZ and LZ B cells. (**E**) Flow cytometric analysis of NP-specific IgG1^+^ B cells (B220^+^Dump [IgG, IgM, Gr-1, CD138, CD3]^-^NIP^+^IgG1^+^) in the spleen. (**F**) The ratio and number of NP-specific IgG1^+^ B cells. (**G**) ELISPOT assay anti-NP IgG1 Ab-secreting plasma cells in the spleen (**H**) Cell number of anti-NP IgG1 Ab-secreting plasma cells. In **B**, **D**, **F**, **H**, each dot represents an individual mouse. All mice were analyzed on day 10 post-immunization with NP-CGG. All data were from two independent experiments. In each statistical analysis, a two-tailed unpaired Student *t*-test was used. *** *p* < 0.001; ns, not significant (*p* > 0.05).

**Figure 5 ijms-23-06584-f005:**
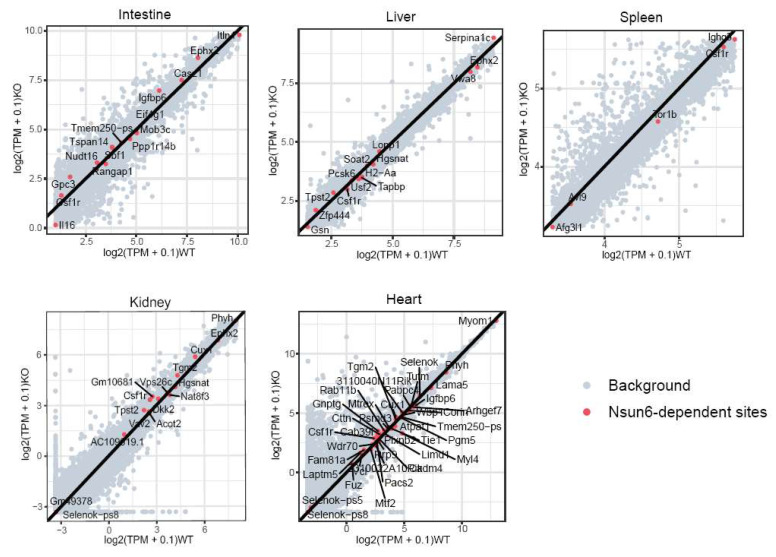
Scatter plots presenting the mRNA level changes of Nsun6 target genes. Scatter plots presenting the mRNA levels of all genes detected in WT and KO tissues. The red dots represent Nsun6 target genes, while the gray dots represent the rest of the genes detected in the tissue. The black line is the diagonal line used as reference.

**Figure 6 ijms-23-06584-f006:**
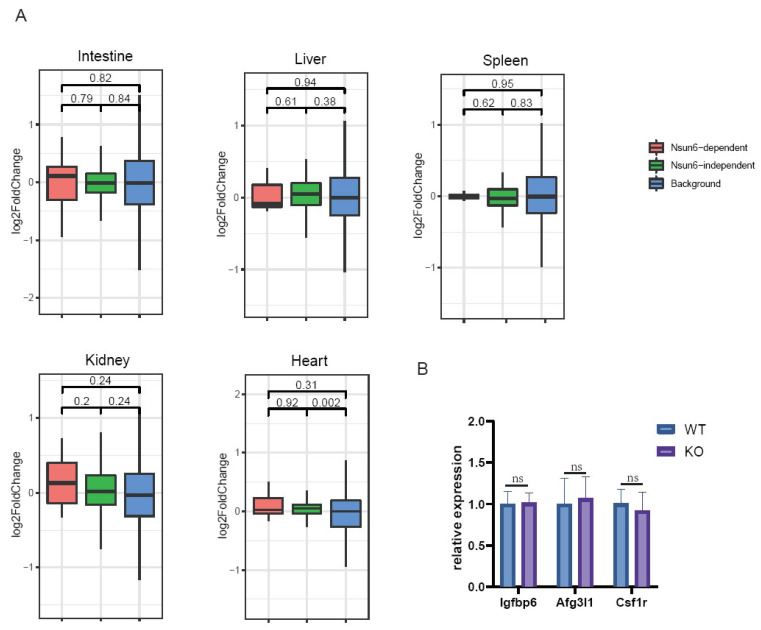
Box plots presenting the global mRNA level changes of Nsun6 target genes. (**A**) The red, green, and blue boxes represent the global mRNA level changes of genes with Nsun6-dependent and -independent m5C sites and of genes without m5C sites, respectively. (**B**) The mRNA levels of highly modified Nsun6 target genes Igfbp6, Afg3l1, and Csf1r were compared between WT and KO heart tissue using RT-qPCR.

## Data Availability

Not applicable.

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
