# Peer review of "A Cross-Tissue Investigation of Molecular Targets and Physiological Functions of Nsun6 Using Knockout Mice"

_ijms, 2022, doi:10.3390/ijms23126584_

Round 1

Reviewer 1 Report

The manuscript describes a rather extensive analysis of the effects of the lack of the Nsun6 5-methylcytosine modification protein. Loss of a related protein, Nsun2, has been shown to have significant physiological effects and the authors hoped that Nsun6 would as well. The short answer is that they showed very little evidence of such an effect. The did demonstrate Nsun6-dependent methylation sites in the mRNAs of various genes and that these modifications are largely tissue specific. But they found no obvious effects of lack of these modifications, notably no effect on mRNA stability. The fact that the gene knockout had no gross effect on the mice, unlike the Nsun2 knockout, suggests that the protein has no strongly significant physiological role, possibly because of redundant effects of one or more of the other Nsun family methylases. The fact that there are some strongly Nsun6 dependent sites doesn't disprove this idea since the other methylases might have subtly different site preferences. My conclusion from reading the manuscript is simply that the largely negative nature of the data could be trivial if there are other substantially redundant methylases. Lacking evidence for or against that idea very much weakens the impact of this work.

Author Response

Reviewer 1

The manuscript describes a rather extensive analysis of the effects of the lack of the Nsun6 5-methylcytosine modification protein. Loss of a related protein, Nsun2, has been shown to have significant physiological effects and the authors hoped that Nsun6 would as well. The short answer is that they showed very little evidence of such an effect. They did demonstrate Nsun6-dependent methylation sites in the mRNAs of various genes and that these modifications are largely tissue specific. But they found no obvious effects of lack of these modifications, notably no effect on mRNA stability. The fact that the gene knockout had no gross effect on the mice, unlike the Nsun2 knockout, suggests that the protein has no strongly significant physiological role, possibly because of redundant effects of one or more of the other Nsun family methylases. The fact that there are some strongly Nsun6 dependent sites doesn't disprove this idea since the other methylases might have subtly different site preferences. My conclusion from reading the manuscript is simply that the largely negative nature of the data could be trivial if there are other substantially redundant methylases. Lacking evidence for or against that idea very much weakens the impact of this work.

Answer:

Thanks for the comprehensive comments.  As for the possibility of redundant effects of other methylases, we and other groups had already proved that m5C modification on mRNAs was deposited either by NSUN2 or NSUN6, and NSUN2 and NSUN6 modified genes are not overlapped [20-22]. Therefore, at least for mRNA, there should be no other methylase that could compensate for the loss of NSUN6. However, as mentioned by Reviewer 1, we cannot exclude the possibility that other functions of Nsun6 might be compensated by other proteins, which awaits further investigation.

Reviewer 2 Report

It is a commendable piece of work that confirms the role of Nsun6 in the overall event of 5-methylcytosine (m5C) modification on mRNA. The study is well presented and supported by appropriate experiments. The set of invivo experiments done in this study clearly facilitate to channelize the direction of future studies for Nsun6. I recommed this work to be of publishable quality in this prestigious journal.

Only a few point that need a bit of attention.

1. A scanning of the article for minor typo errors (for example, line 121: ..Then, we toke union .., line 310: understanding of Nsun6’s phycological functions)  and unification of the font (example line 70) must be done.  

2. Since the results concludes that the only function of Nsun6 is in the development of Ab secreating plasma cells, the study will benfit from a targetted discussion of the role of Nsun6 in the the formation of antibody-secreting plasma cells in the light of previous studies. Furthermore, the data presented in figure 3 (C and D) should be integrated in the discussion. 

Author Response

Reviewer 2

It is a commendable piece of work that confirms the role of Nsun6 in the overall event of 5-methylcytosine (m5C) modification on mRNA. The study is well presented and supported by appropriate experiments. The set of in vivo experiments done in this study clearly facilitate to channelize the direction of future studies for Nsun6. I recommend this work to be of publishable quality in this prestigious journal.

Only a few point that need a bit of attention.

  1. A scanning of the article for minor typo errors (for example, line 121: ..Then, we toke union .., line 310: understanding of Nsun6’s phycological functions)  and unification of the font (example line 70) must be done.  

Answers:

Thanks for your careful reading. We have carefully revised our text and corrected these typos.

  1. Since the results concludes that the only function of Nsun6 is in the development of Ab secreting plasma cells, the study will benefit from a targeted discussion of the role of Nsun6 in the formation of antibody-secreting plasma cells in the light of previous studies. Furthermore, the data presented in figure 3 (C and D) should be integrated in the discussion. 

Answers:

Thanks for your constructive suggestion. We have discussed the result in figure 3C&D and added a targeted discussion of the possible role of Nsun6 in the formation of antibody-secreting plasma cells in the second paragraph of the discussion (Line 270-280).

Reviewer 3 Report

The aim is stated clear. The authors stated clearly what study found and how they did it.

The title is informative and relevant.

The references are relevant and recent. The cited sources are referenced correctly. Appropriate and key studies are included.

The introduction reveals what is already known about this topic. The research question is clearly outlined. The research question also justified given what is already known about the topic.

The process of selection of the subjects was clear. The variables are well defined and measured appropriately. The study methods are valid and reliable. There are enough details provided in order to replicate the study.

The data is presented in an appropriate way. The text in the results add to the data and it is not repetitive. Statistically significant results are clear. It is clear which results are with practical meaning. Results are discussed from different angles and placed into context without being overinterpreted.

The conclusions answer the aim of the study. The conclusions are supported by references and own results.

The limitations of the study are not fatal, but they are opportunities to inform future research.

Specific comments on weaknesses of the article and what could be improved:

Major points - none   

Minor points

1.           Please, state the limitations of the study

2.           Could you please discuss the clinical implications of the results

Author Response

Minor points

  1. Please, state the limitations of the study.

Answers:

Thanks for the suggestion. Our study only analyzed the RNA stability in Nsun6 KO tissues. Although data from HAP1 cells suggested that Nsun6 KO is not likely to affect the transportation and translation of its target genes, we still lack conclusive data to prove this point in mouse tissues, which we state at the end of the 4th paragraph of the discussion.

  1. Could you please discuss the clinical implications of the results.

Answers:

Thanks for the suggestion. Our data demonstrated that Nsun6 KO does not affect the development of the mouse, and, therefore, should not be associated with genetic disease. Based on the phenotypes we observed in antibody immune response, we could only speculate that it may be involved in some immune-related diseases, which requires further investigation. To explore other potential involvement of Nsun6 in diseases, we should perform a more comprehensive analysis of the KO mice to understand whether Nsun6 is involved in other critical biological processes.

Round 2

Reviewer 1 Report

My assessment of the manuscript is unchanged. The fact that the loss of Nsun6 has no physiological effect in the mouse, unlike Nsun2, suggests that either its physiological effect is minimal or that it is redundant with one or more other Nsun proteins. The authors mention that Nsun6 appears to be biochemically distinct, in the sense that it has apparently unique methylation sites in the genome but this does not answer the question whether those sites are themselves essential for some physiological effect. If that were the case then one would expect the knockout of Nsun6 to have a measurable physiological effect, which it doesn't. That suggests that the specific Nsun6 methylation sites are not essential for whatever role the protein plays in the cell. It remains true that the authors should consider looking for synthetic phenotypes between the Nsun6 knockout mutant and mutants of other Nsun protein genes. For example, the lack of Nsun6 might intensify the phenotype resulting from the lack of the Nsun2 protein. Or, the mice lacking Nsun6 and another apparently inessential Nsun protein might display a novel phenotype.